# A Novel Inorganic Aluminum Phosphate-Based Flame Retardant and Thermal Insulation Coating and Performance Analysis

**DOI:** 10.3390/ma16134498

**Published:** 2023-06-21

**Authors:** Guoshuai Cai, Jiaxuan Wu, Jiayi Guo, Yange Wan, Qingjun Zhou, Pengyu Zhang, Xiaolei Yu, Mingchao Wang

**Affiliations:** 1School of Safety Science and Engineering, Civil Aviation University of China, 2898 Jinbei Road, Tianjin 300300, China; caiguoshuai0820@163.com (G.C.); guojy_0803@163.com (J.G.); wanyange2022@163.com (Y.W.); 2College of Science, Civil Aviation University of China, 2898 Jinbei Road, Tianjin 300300, China; wjx2534413953@163.com (J.W.); qjzhou@cauc.edu.cn (Q.Z.); 3Tianjin Building Materials Science Research Institute Co., Ltd., Tianjin 300381, China; zpy329@163.com; 4Dezhou Zhongke New Materials Co., Ltd., Dezhou 253011, China

**Keywords:** flame-retardant coating, thermal insulation, composition and structural evolution, thermal property, bonding performance

## Abstract

Currently, most thin-layer expandable coatings are polymer-based, with very few inorganic expandable coatings. Due to the high environmental friendliness of inorganic coatings, studying new types of inorganic coatings is of great significance. A novel amorphous aluminum phosphate-based flame-retardant coating was prepared by modifying it with nano-silica, hollow silica beads, hollow glass microspheres, and boron carbide. A comprehensive study was conducted on the flame retardancy and thermal insulation performance, composition and structural evolution under flame and physical and chemical properties, and the mechanisms of flame retardancy and thermal insulation were elucidated. Large-plate combustion testing, bonding strength testing, XRD, IR, TG-DSC, and SEM testing were all applied in this work. The synergistic effect of the four fillers was very obvious, and a series of AP22XY (nano-silica/silica beads/hollow glass microspheres/boron carbide = 2:2:0:4, 2:2:1:3, 2:2:2:2, 2:2:3:1, 2:2:4:0) coatings were prepared. The change in the ratio of glass microspheres to boron carbide had a significant impact on the composition and structural evolution of the coating, thus reflecting its effectiveness as a flame retardant and thermal insulation. Although decreasing the ratio would promote the formation of borosilicate glass and Al_18_B_4_O_33_ and improve the thermal stability of coatings, the structure inside of the coating, especially the skeleton, would be dense, which is not conducive to thermal insulation. When the ratio of glass microspheres to boron carbide is 3:1, AP2231 shows the best fire resistance. Under the combustion of butane flame at about 1200–1300 °C, the backside temperature reaches a maximum of 226 °C at 10 min, and then the temperature gradually decreases to 175 °C at 60 min. This excellent performance is mainly attributed to three aspects: (1) the foaming and expandability of coatings when exposed to fire, (2) the multiple endothermic reactions the coating undergoes, and (3) the improvement effect of boron carbide. Additionally, AP2231 shows the best bonding performance with a strength of close to 4.5 MPa after combustion, because of the appropriate content matching between borosilicate glass, Al_18_B_4_O_33_, and hollow glass microspheres. The coating has potential application prospects in the construction and transportation fields, such as the protection of structural steel, fire prevention in subways and tunnels, and the prevention of lithium battery fires.

## 1. Introduction

Fire is an essential part of daily human life, driving the development and progress of human society. If an accident occurs during the use of fire, causing the flame to become uncontrollable, it will evolve into an accident. Fire accidents damage the achievements of human history and civilization, affect social health and stability, and hinder the rapid development of the social economy [1,2]. With the continuous development of society, there are more ways in which humans use fire, leading to an increasing number of fire accidents. According to statistics, about 165,000 people die from fires every year, so fire prevention is very significant [3]. In addition to actively eliminating fire hazards and preventing fires, it is also important to systematically research and develop fire prevention modes.

Currently, there are mainly three fire prevention modes: sensor warning, active extinguishing with fire-extinguishing agents, and passive fire prevention [4,5,6]. The sensor early-warning system adopts a physical method of non-contact temperature measurement, combined with a fire alarm algorithm built into the monitoring front-end hardware for intelligent judgment and a timely alarm, to prevent imminent fire crises [7,8]. Active extinguishing refers to the use of fire-extinguishing agents to prevent combustible gases, liquids, and solids from coming into contact with combustion aids such as air, oxygen, or other oxidizing agents, thereby achieving a flame-retardant effect [9,10]. Passive fire prevention is a method of improving or utilizing the strong fire resistance of materials, slowing down the speed of flame spread or preventing combustion within a certain period of time, including both fully flame-retardant material and flame-retardant coating methods [11,12,13]. Due to the wide variety of materials, it is not possible to prepare all materials as flame retardants from a technical perspective. Therefore, the application of flame-retardant and thermal insulation coatings on the surface of materials has become a hot research topic. 

According to the fire prevention mechanism, flame-retardant coatings can be divided into non-intumescent coatings and intumescent coatings [14,15]. Non-intumescent flame-retardant coatings are generally thick-coated coatings. They usually achieve the function of fire prevention and thermal insulation by utilizing the low thermal conductivity and noncombustibility of the material itself [16]. Non-intumescent coatings have excellent characteristics, such as a long fire resistance time; are non-expansive, non-combustible, and non-explosive; and do not release harmful gases in the event of a fire [17]. However, this coating is usually thicker, more expensive, and heavier; thus, it is commonly used in practical applications for increasing fire protection. Intumescent flame-retardant coatings are a more widely used type of coating usually prepared by using water or organic solvents as dispersants, polymer resins or emulsions as film-forming agents, and are modified with foaming agents, flame retardants, carbon-forming agents, and other additives [18,19,20]. The coating can vigorously foam to form a refractory honeycomb or sponge-like insulation carbon layer when encountering flames. The residual carbon structure itself has certain thermal insulation characteristics [21]. Meanwhile, during the foaming or carbonization process, some non-flammable gases will be released to dilute oxygen to some extent and prevent the expansion of the fire. Although this type of coating has extremely decorative properties and excellent flame retardancy and thermal insulation properties, the use of organic compounds still results in high smoke emissions after they encounter fire, which is not conducive to environmental protection. 

Due to its excellent high-temperature resistance and adhesion/binding properties, aluminum phosphate is applied for the manufacture of refractory ceramic materials, sealing of thermally sprayed coatings, and ceramic coatings themselves [22,23]. Formanek et al. [23] prepared a thermal-protective coating using aluminum phosphate binders mixed with oxides and ceramic powders, which showed excellent high-temperature corrosion resistance. However, most aluminum phosphate binders are prepared with a higher P/Al ratio, usually P/Al = 3, which leads to a highly crystallized form or a geopolymer state. So, it is hard to develop an expandable coating. In recent years, a kind of amorphous aluminum phosphate has been developed to prepare heat-resistant adhesives and lightweight thermal-insulated porous ceramic [24,25,26,27,28,29], which is a special type of “inorganic resin” with a wide network structure. It will also decompose, foam, and expand when exposed to fire. Although its foaming and expansion effect is not as effective as that of organic polymers, the high heat resistance and low thermal conductivity of the decomposition products lend it a certain thermal insulation effect [30]. The most important thing is that its decomposition product is mainly water, with no toxic substances released [31]. The study provides a new idea for the preparation of new inorganic expandable flame-retardant coatings. Additionally, amorphous aluminum phosphate has an extremely high bonding strength, which can enhance the practicality of the coatings. Wang et al. [3] prepared an inorganic expandable coating using an amorphous aluminum phosphate-based emulsion as the matrix, and ammonium phosphate tetrahydrate, nitric acid, and oxalic acid as foaming agents. Under a butane flame, the coating was completely noncombustible and the back temperature was only 140 °C after 60 min, demonstrating its excellent insulation performance. However, due to the addition of foaming agents, nitrogen-containing smoke (N_2_O, NO_2_, etc.) was released upon exposure to fire, and the smoke density reached 64.73%. Its further application is greatly limited by smoke emissions. Therefore, it is of great significance to utilize non-decomposable inorganic components (such as nano-silica, hollow silica beads, hollow glass microspheres, and boron carbide) to improve the structure of amorphous aluminum phosphate-based coatings and prepare new environmentally friendly and expandable inorganic coatings. 

Nano-silica has excellent mechanical strengthening properties and low thermal conductivity, and hollow silica microsphere has better insulation properties. The synergy between them satisfies the optimization of particle size distribution, as the mechanical properties of the coating and the heat transfer mechanism of the coating structure can be improved by adjusting the proportions [32,33]. In addition, they are often used as reaction sources for ceramic or glass phases at elevated temperatures. The introduction of glass microspheres with larger sizes has the greatest impact on the coating structure, thereby affecting the insulation effect of the coating [34]. Simply adding glass microspheres can easily introduce more defects inside the coating, leading to thermal cracking and loss of insulation. Therefore, the synergistic addition of small-sized silica and large-sized glass microspheres is beneficial for reducing internal defects in the coating, improving the structural integrity of the coating, and ensuring a significant insulation effect. The oxidation of boron carbide will result in 2.5-times-greater volume, and the generated molten boron oxide can significantly improve the coating structure [35]. The single addition of boron carbide has two main drawbacks. Firstly, the oxidation of boron carbide is an exothermic reaction, which will increase the temperature of the back of the coating; secondly, the generated boron oxide is prone to volatilization due to its inability to participate in further reactions on time, which can easily cause damage to the coating structure. Generally, the amount of boron carbide added is small, and it needs to be co-added with silica to generate borosilicate glass through a high-temperature reaction to retain it. Therefore, the effect of adding these fillers alone is limited, and the synergistic effect between them is worth studying. However, the single and synergistic effects of the four fillers mentioned above have rarely been studied. In this work, to investigate the effect of various fillers on the flame retardancy and thermal insulation performance of coatings, the effects of adding single fillers, double fillers, and multiple fillers on flame retardancy were compared, and the optimal formula was determined. The composition and structure evolution, as well as the thermal property of the coatings, were comprehensively studied to explore the flame retardancy and thermal insulation mechanism.

## 2. Materials and Methods

All raw materials were used directly without further processing. Phosphoric acid (85 wt.%) and aluminum hydroxide (analytical purity) were purchased from Aladdin Reagent Co., Ltd., Shanghai, China. Nano-silica powder (d < 100 nm) and hollow mesoporous silica beads (d = 900 nm) were provided by Mingchuang Innovation Materials Co., Ltd., Xiangtan, China. Hollow glass microspheres (d = 10 μm) were provided by Jingong Sili glass beads Co., Ltd., Tiantai, China. Boron carbide (d = 5 μm) was purchased from Zhengxing Abrasive Co., Ltd., Dunhua, China. 

The concentrated phosphoric acid was first diluted to 50% at room temperature, and then it was heated to 85 °C using a water bath. Aluminum hydroxide powders were slowly added to the heated dilute phosphoric acid (Al/P = 1.2:1) under mechanical stirring at 300–500 r/min, and the coating base solution was obtained when the viscosity reached 1000–1500 mPa·s. Secondly, nano-silica powder, hollow silica beads, hollow glass microspheres, and boron carbide powders were added to the base solution according to the formula shown in Table 1. The slurry was obtained after mechanical stirring at 500–700 r/min for 5 h. The prepared coatings were stored in tanks for standby. 

Two kinds of steel plates (100 mm × 100 mm × 1 mm and 40 mm × 40 mm × 1 mm) were prepared to utilize wire cutting. The large plates were used for the large-panel combustion test, and the small plates were used for the interfacial bonding test. Before coating, these plates were first polished with # 800 and # 1500 sandpapers and cleaned with alcohol and stored until use. According to the coating process introduced in Ref. [3], the thickness of the cured coating was controlled at ~2 mm by using a coating applicator. After curing in the atmospheric environment with a humidity of 60–80% for 36 h, the post-tests could be carried out.

The flame retardancy and thermal insulation performance of different coatings were tested by using the large-panel combustion method with a butane flame burner. The temperature of the flame was approximately 1200–1300 °C, and the backside temperature behind coatings was collected using a real-time data acquisition device. The horizontal distance between the sample and the nozzle of the butane flame burner was 8 cm throughout the test. Five temperature sensors were used for each test, and each formula was tested three times. The highest temperature curve for the three tests was averaged. The interfacial bonding strength between adhesive and steel plates before and after combustion was tested by using a CMT4504 testing machine according to Standard ASTM C633-2001. E51 epoxy was applied to prepare tensile parts. Before that, the outermost coating was removed first, leaving only the bottom coating. 

The composition of different coatings under different conditions was tested using a D/Max-2500 X-ray diffractometer (XRD, Rigaku, Tokyo, Japan) and WQF-530 Fourier infrared analyzer (FTIR, Beifen-Ruili, Beijing, China). TG-DSC curves of different coatings were tested using an STA-449C synchronous thermal analyzer (Netzsch, Gerätebau, Bavaria, Germany) with air flowing at a heating rate of 20 °C/min. Finally, the surface and cross-sectional micromorphology of coatings after combustion were observed using an S-4800 scanning electron microscope (Hitachi, Tokyo, Japan). 

## 3. Results and Discussion

### 3.1. Fire Resistance Performance

Figure 1a shows the fire resistance performance of the AP coating modified with only one additive. When only boron carbide was added, the flame retardancy and thermal insulation effect were the worst. The backside temperature of AP0008 exceeded 220 °C by 70 s and reached almost 370 °C at 500 s. From 500 °C to 2000 °C, there was a slight decrease in temperature. Then, the temperature gradually increased with time and remained at about 335 °C at 3600 s. The effect of simply adding glass microspheres to the coating (AP0080) was not very good either. Its backside temperature also reached 220 °C by 90 s and reached 310 °C at 500 s. Then, it remained in a descending state from 500 s to 3600 s, and finally decreased to 237 °C. By comparison, the thermal insulation effect of nano-silica and silica microspheres was better. In particular, adding silica microspheres to the coating (AP0800) led to a better performance in the first 2000 s than adding nano-silica (AP8000). The back temperature of AP0800 reached 220°C at 376 s, while AP8000 advanced by about 100 s. The maximum temperature on the back of the two coatings was close, with both being around 260°C. However, AP8000′s thermal insulation effect was superior to that of AP0800 from 2000 to 3600 s, and its backside temperature was 22 °C lower than that of AP0800. Additionally, the bonding strength of various coatings on steel before and after combustion is shown in Figure 1b. Before combustion, the enhancement effect of the four fillers on the bonding strength of the coating followed the order, from strong to weak: boron carbide, nano-silica, silica microspheres, and glass microspheres. Although the addition of boron carbide alone could increase the room temperature bonding strength of the coating to 7.2 MPa, the strength of the coating sharply decreased to less than 1 MPa. This result is ascribed to the boron oxide generated by the oxidation of boron carbide, which does not participate in further reactions and evaporates under the flame [31]. As seen in Figure 1f, there were many large pores forming on the surface of the AP0008 coating. Regarding the AP0080 coating, the bonding performance was not good in either case, which may have been due to the strengthening effect of larger spheres. Additionally, due to the insufficient temperature-bearing capacity of the glass microspheres themselves, the coating also had many surface pores after calcination, as shown in Figure 1e. For silica-enhanced coatings, there were no pores on the surface of the coating, although some cracks were observed. The surface cracks of AP8000 were few but large, while the surface cracks of AP0800 were very small turtle cracks.

Figure 2 shows the effect of the addition of two fillers on the flame retardancy performance of the coatings. The additional effect of dual components is relatively better than that of a single component. On the other hand, the combination of glass microspheres and boron carbide had the worst effect, and the maximum backside temperature of AP0044 reached 303 °C at 500 s. Second, the combination of nano-silica and boron carbide worked slightly better; the maximum temperature behind the AP4004 coating did not exceed 280 °C (~480 s). The temperature continued to decrease over time and dropped below 220 °C after 60 min. By comparison, the combination of nano-silica and silica microspheres showed the best effect. The AP4400 coating maintained a temperature of 220 °C for 620 s, and the back temperature could be continuously controlled at around 250 °C. For bonding performance, the combination of nano-silica and boron carbide worked best, and the bonding strength before and after combustion was 7.62 MPa and 3.45 MPa, respectively. The bonding effect of coatings enhanced with glass microspheres was still not very good, especially since the coordination between microspheres and boron carbide was the worst. Regarding the surface state after combustion, the synergistic participation of nano-silica and silica microspheres prevented the generation of a large number of microcracks (see Figure 2c), and the AP4400 coating had the highest surface morphology integrity. Two-component-modified coatings with boron carbide or glass microspheres all exhibited pores or cracks on the surfaces. Generally, adding excessive boron carbide or glass microspheres can easily introduce pores (see Figure 2d,e), while adding silica microspheres can easily introduce cracks (see Figure 2f).

Since the addition of both nano-silica and silica microspheres can effectively block the growth of surface cracks when their mass ratios are the same, we further investigated the effect of different ratios of glass microspheres and boron carbide on the properties of coatings. It is found that the synergistic addition of four additives can effectively enhance the flame retardancy and thermal insulation effect of the coating. Figure 3 shows the fire resistance performance of the AP coating modified with four additives. On the basis of adding nano-silica or silica microspheres of the same quality, adding only boron carbide (AP2204) or glass microspheres (AP2240) alone could not achieve the best effect. When glass microspheres and boron carbide were added in a 3:1 mass ratio, the coating exhibited the best thermal insulation performance. The maximum temperature behind AP2231 was only 226 °C at 600 s, and then the temperature gradually decreased to about 175 °C at 3600 s. Additionally, although the insulation effect of the AP2222 coating was better than that of AP2231 in the long term, the temperature behind it was higher at around 500 s. Therefore, AP2231 is the best formula. 

Figure 3b shows the effect of different ratios of glass microspheres and boron carbide on the coatings’ adhesion properties. With the addition of four fillers, the bonding effect of the coating was significantly improved, due to the synergistic strengthening effect brought about by the combination of different sizes of fillers. The bonding strength before and after combustion shows a trend of first increasing and then decreasing with the increase in the ratio. Before combustion, the bonding strength of AP2213 coating reached a maximum value of 8.4 MPa; after combustion, AP2231 showed the best bonding effect with a strength of close to 4.5 MPa. 

Figure 3c,d show photos of coatings modified with four additives during and after combustion. AP2222 had the best integrity; even after 1 h of combustion, there were no cracks or pores on the surface. Next was AP2213; only a small number of cracks appeared after 1 h of combustion. For AP2204 and AP2240, there were still many holes on the surface of these two coatings. Finally, the AP2231 coating had a poor appearance, so its ability to provide good thermal insulation performance is also related to its component and structural evolution with temperature. In the subsequent narrative and discussion, a detailed characterization of these five coatings has been conducted to explore their flame retardancy and thermal insulation mechanisms.

### 3.2. Component Analysis

Figure 4a shows the XRD spectra of the char of different AP22XY coatings (nano-silica/silica beads/ glass microspheres/boron carbide = 2:2:0:4, 2:2:1:3, 2:2:2:2, 2:2:3:1, 2:2:4:0) after 1 h of combustion. There was still residual Al(OH)_3_ in AP2213, AP2222, and AP2231, indicating that these coatings themselves have good lateral thermal insulation, and the coating exposed to the edge of the flame was not affected. AlPO_4_, Al_18_B_4_O_33_, BPO_4_, and SiO_2_ are the main ceramic components of these chars formed under fire. Additionally, there was some boron-containing glass formed in these coatings, as is evidenced by the IR absorption peak of B-O vibration at 1400–1500 cm^−1^ in Figure 5a [36]. Due to the uneven heating of the coating during the flame-burning process, using residual char from the coating to compare the differences in the components of different coatings is not accurate enough. Therefore, the composition of these coatings after calcination in a furnace at 1200 °C was determined, and their XRD and IR results are shown in Figure 4b and Figure 5b, respectively. Compared to Figure 4a,b, the main crystallization peaks appear in both figures, which indirectly indicates that the temperature of the flame can reach 1200 °C. 

After calcination at 1200 °C, the crystal form of aluminum phosphate was entirely orthorhombic. With the increase in B_4_C content, the crystal peaks of silica and alumina gradually decreased or even disappeared, which indicates the formation of borosilicate glass and Al_18_B_4_O_33_, and the content of Al_18_B_4_O_33_ gradually increased from AP2240 to AP2204. As the glass microspheres are also a component of borosilicate glass, Al_18_B_4_O_33_ and silica were not generated in AP2240, indicating that glass microspheres have high heat resistance and do not undergo crystallization. As seen in Figure 5b, the intensity of the B-O absorption peak at 1400–1500 cm^−1^ also increased with increasing B_4_C, which further proves the increase in borosilicate glass content. Therefore, the main components of AP2222 and AP2231 coatings at the front edge of the fire were borosilicate glass, Al_18_B_4_O_33_, and AlPO_4_. 

To further explore the composition evolution of the AP2231 coating with temperature, its XRD and IR spectra were collected after calcination at different temperatures, and are shown in Figure 4c and Figure 5c, respectively. At room temperature, only Al(OH)_3_ was identified. Now, aluminum phosphate exists in the form of amorphous macromolecules, known as inorganic resin containing a large number of hydroxyl groups and bound water [36,37]. The peaks of the fillers were not visible, partly because the microspheres were all amorphous and partly because their content was low. Moreover, after being coated with macromolecules, they were less easily identified on the surface by XRD. With the temperature increasing, the macromolecular structure began to decompose, and hexagonal AlPO_4_ appeared at 400 °C. The intensity of the infrared absorption peak of -OH at 1650 cm^−1^ sharply decreased from RT to 400 °C (see Figure 5c), also indicating that the decomposition process is accompanied by a loss of moisture. Additionally, most of Al(OH)_3_ decomposed to produce AlO(OH). After calcination at 800 °C, the crystal intensity of hexagonal AlPO_4_ further increased, and hexagonal Al(PO_3_)_3_ appeared, which implies that there are still some hydrated aluminum phosphates existing in the range of 400 °C to 800 °C. The loss of bound water results in the generation of more crystal aluminum phosphates. Meanwhile, the B-O absorption peak (1400–1500 cm^−1^) began to appear at 800 °C, indicating that the oxidation of B_4_C occurs above 600 °C. In addition, AlO(OH) completely decomposed at 800 °C, but there were no crystal peaks of Al_2_O_3_ present; it can thus be concluded that the decomposed Al_2_O_3_ is amorphous (B_2_O_3_ also). 

From 800 °C to 1000 °C, the evolution of the crystal form mainly occurred within the coating. In particular, Al(PO_3_)_3_ transformed from the hexagonal phase to the cubic phase, and AlPO_4_ transformed from the hexagonal phase to the orthorhombic phase. As the temperature further increased, cubic Al(PO_3_)_3_ further reacted with Al_2_O_3_ to produce orthorhombic AlPO_4_, and B_2_O_3_ further reacted with Al_2_O_3_ to produce Al_18_B_4_O_33_ at 1200 °C. Finally, the main stabilizing products of the AP2231 coating were orthorhombic AlPO_4_, Al_18_B_4_O_33_, and borosilicate glass. Additionally, a slight decrease occurred in the IR peak of B-O at 1200 °C (see Figure 5c), indicating that a certain amount of boron oxide had evaporated.

### 3.3. Thermal Properties

Figure 6 shows the TG-DSC curves of AP2204, AP2222, AP2231, and AP2240 from RT to 1200 °C. The mass loss of these coatings in the temperature range of RT −600 °C can be divided into four stages. The first stage occurs in the range of 100-140°C, representing the further condensation of phosphate and the volatilization of free water. The mass loss in the second stage is similar, which is likely due to the decomposition of Al(OH)_3_ to AlO(OH). The third stage around 300 °C is the most severe stage of phosphate decomposition. According to the component analysis mentioned above, the decomposition products are AlPO4, Al(PO_3_)_3_, and partially hydrated phosphates, and they maintain high stability until 450 °C. The fourth stage occurs in the range of 450 °C–600 °C, which corresponds to the generation of more AlPO_4_ decomposition products. The mass loss in the first stage is affected by the ratio of glass microspheres to boron carbide. As the ratio increased, the mass loss increased from 12.7% for AP2204 to 18.7% for AP2240. Also, the total mass loss rate also follows this trend: the higher the content of B_4_C, the lower the mass loss. This result indicates that boron carbide can enhance the thermal stability of coatings and improve the mass of coatings during combustion. As is well known, the oxidization of B_4_C to B_2_O_3_ could bring about mass improvement and volume increase. Meanwhile, the oxidation reaction releases heat. Therefore, the DSC baseline of the coating modified with B_4_C was higher. Of course, the overall increase in the DSC curve also comes from the transformation of various crystal phases of aluminum phosphates to stable orthorhombic AlPO_4_. 

On the other hand, when the content of B_4_C is higher, the endothermic peaks will clearly appear on the DSC curve, as seen in Figure 6a,b. The first endothermic peak around 900–950 °C likely results from the formation of borosilicate glass, and the second peak around 1050 °C is due to the volatilization of excess boron oxide. The decrease in the TG curve around 1000 °C further demonstrates this volatilization process. The thermal results mentioned above are consistent with the component analysis.

### 3.4. Structure Analysis

In addition to component evolution, structural evolution also deeply affects the flame retardancy and thermal insulation effect. The surface morphology of AP2204, AP2213, AP2222, AP2231, and AP2240 after the flame exposure is shown in Figure 7. As seen in the low-magnification image shown in Figure 7a1,b1,c1,d1,e1, with an increasing ratio of glass microspheres to boron carbide, the density and integrity of the coating surface became worse and worse. In particular, when there are more glass microspheres, larger holes will appear on the surface. As seen in Figure 7e1, some deep holes with a size of about 300 μm are clearly visible on the surface of the AP2240 coating. The size and number of these pores become smaller and smaller as B_4_C content increases, and almost no holes appear on the surface of AP2204 (see Figure 7a1), which is ascribed to the effective volume compensation caused by the oxidization of B_4_C. As seen in Figure 7a2,a3, the microstructure of the coating was still very dense, and the surface was like a flowing liquid. The solidified liquid was likely the excess B_2_O_3_. As the content of glass microspheres increased, their arrangement became increasingly dense from AP2213 to AP2240. These glass microspheres were still in good condition after being exposed to fire, and the damage was relatively minor. Meanwhile, for coatings with boron carbide added simultaneously, the surface of glass microspheres usually adhered to other substances formed by the transition from the liquid to solid state. It is inferred that they are borosilicate glass. Therefore, it can be deduced that the generated borosilicate glass also provides some protection for the glass microspheres.

Figure 8 shows the cross-sectional morphology of various coatings after combustion. After exposure to fire, there were many internal pores in the coatings, including large pores between the residual carbon skeleton and small pores on the skeleton, as seen in Figure 8. The generation of these pores comes from the decomposition of the coating matrix. Of course, their size will be regulated by the fillers. As the ratio of glass microspheres to boron carbide increased, the size of both large pores and small pores increased. The size of large pores increased from ~300 μm in AP2204 to over 1 mm in AP2240. The skeleton was relatively dense in AP2204 and AP2213 coatings, with no obvious small holes visible on it. The dense skeleton structure was not conducive to thermal insulation, although the surfaces of both coatings were very dense. By comparison, many small, closed pores appeared on the skeletons of AP2222, AP2231, and AP2240. Overall, the size of these micropores increased with decreasing boron carbide. These frameworks filled with closed pores are very beneficial for thermal insulation, which is one of the main reasons why AP2231 exhibited an excellent flame retardancy performance.

### 3.5. Flame Retardancy and Thermal Insulation Mechanism 

As all the materials used for preparing coatings are incombustible inorganic compounds, and the products are all ceramic phases, when the coating reacts with fire, the coating itself is completely noncombustible. The biggest advantage of this coating is its insulation performance. Here, AP2231 was used as an example to illustrate an excellent thermal insulation performance. It can be summarized in the following three points. 

Firstly, the coatings exhibited good foaming and expandability performances when exposed to fire. The amorphous aluminum phosphate matrix is a macromolecular structure similar to the polymer. When it is heated, it decomposes and generates a large number of pores, resulting in a certain thickness expansion. This not only improves the thermal insulation of the matrix but also provides a foundation for further optimization of the structure.

Secondly, the coating has multiple endothermic reactions. With the introduction of a flame, multiple endothermic reactions occur in the coating, including the decomposition of phosphate (140–300 °C), the decomposition of aluminum hydroxide (400–600 °C), the generation of glass (900–950 °C), and the volatilization of partial boron oxide (~1050 °C). These reactions prevent an increase in heat by consuming a certain flame temperature. 

Thirdly, the improvement effect of boron carbide is significant. The oxidization of B_4_C leads to an expansion of about 2.5 times in volume and generates molten B_2_O_3_. The liquid B_2_O_3_ can not only repair cracks but also improve the closed pore structure. Moreover, boron oxide will further react with silica and alumina to form borosilicate glass and Al_18_B_4_O_33_, respectively, thus improving the fire resistance of the coating.

In addition, unlike the third mechanism, the first two mechanisms are applicable to all coatings. The above processes do not occur immediately after exposure to fire, and they require a certain amount of time. As seen in Figure 1, Figure 2 and Figure 3, the shapes of these curves seem to indicate growth up to a maximum value and then a decrease, followed by stabilization. The initial temperature increase is because the coating has not yet fully expanded, and the heat transfer rate is high. With the extension of time, the coating structure and components undergo higher evolution, showing higher insulation and lower thermal conductivity. Therefore, the temperature on the back will gradually decrease or stabilize. This phenomenon is more pronounced for coatings with added boron carbide, as the oxidation of boron carbide releases heat.

## 4. Conclusions

A novel inorganic flame-retardant coating was prepared with an amorphous aluminum phosphate-based emulsion as the matrix, and nano-silica, hollow silica beads, hollow glass microspheres, and boron carbide as fillers. The effects of adding single fillers, double fillers, and multiple fillers on flame retardancy were compared. Large-plate combustion testing, bonding strength testing, and XRD, IR, TG-DSC, and SEM testing were all applied to analyze the performance and physicochemical properties of coatings. 

For coatings with only one filler added, the back temperature of the coating was higher than 250 °C, especially for coatings with only hollow glass microspheres added, for which the back temperature could even reach above 350 °C due to the generation of numerous cracks and pores under a flame. The performance of the coating modified with double fillers improved to some extent, but it was not significant enough. In contrast, the synergistic effect of the four fillers was very obvious. 

A series of AP22XY coatings with a filler ratio of 2:2:X:Y (nano-silica/silica beads/glass microspheres/boron carbide = 2:2:0:4, 2:2:1:3, 2:2:2:2, 2:2:3:1, 2:2:4:0) were studied and compared. Although decreasing the ratio would promote the formation of borosilicate glass and Al_18_B_4_O_33_ and improve the thermal stability of the coating, the structure inside of the coating, especially the skeleton, would be denser, which is not conducive to thermal insulation. When the ratio of glass microspheres to boron carbide is 3:1, AP2231 showed the best fire resistance performance. Under the combustion of butane flame at about 1200 °C–1300 °C, the backside temperature reached a maximum of 226 °C at 10 min, and then the temperature gradually decreased to 175 °C at 60 min. This excellent performance is mainly attributed to three aspects: (1) the foaming and expandability of coatings when exposed to fire, (2) the coating’s multiple endothermic reactions, and (3) the improvement effect of boron carbide. Additionally, AP2231 showed the best bonding performance with a strength of close to 4.5 MPa after combustion, because of the appropriate content matching between borosilicate glass, Al_18_B_4_O_33_, and hollow glass microspheres. 

The phosphate coatings in this work were used in room-temperature environments, especially in areas with slight dampness. They can play a role in fire prevention and have potential applications in the protection of structural steel, fire prevention in subways and tunnels, the prevention of lithium battery fires, and other areas.

## Figures and Tables

**Figure 1 materials-16-04498-f001:**
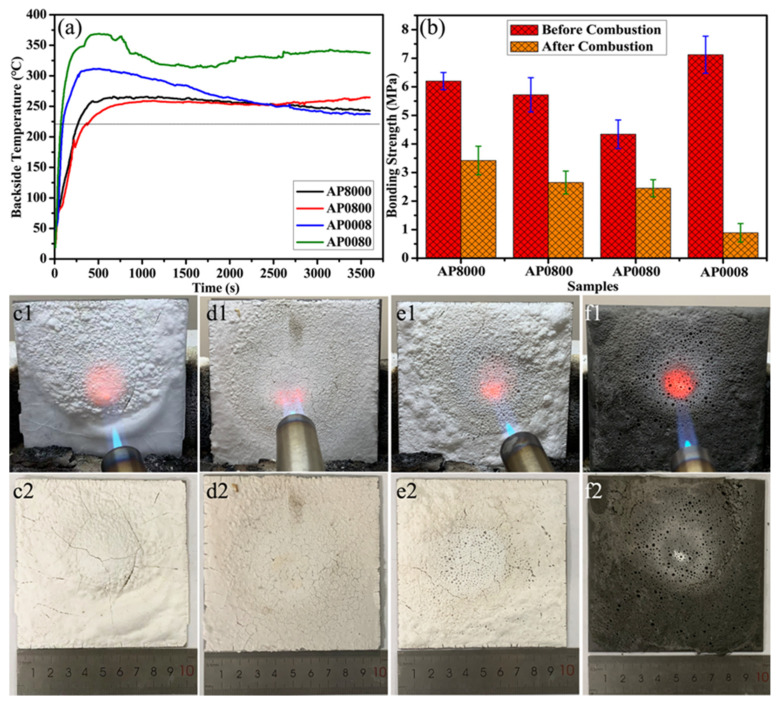
The fire resistance performance of AP8000, AP0800, AP0080, and AP0008: (**a**) the backside temperature under butane flame, (**b**) the comparison of bonding strength before and after combustion, and photos of the coatings (**c1**: AP8000 during combustion, **d1**: AP0800 during combustion, **e1**: AP0080 during combustion, **f1**: AP0008 during combustion, **c2**: AP8000 after combustion, **d2**: AP0800 after combustion, **e2**: AP0080 after combustion, **f2**: AP0008 after combustion).

**Figure 2 materials-16-04498-f002:**
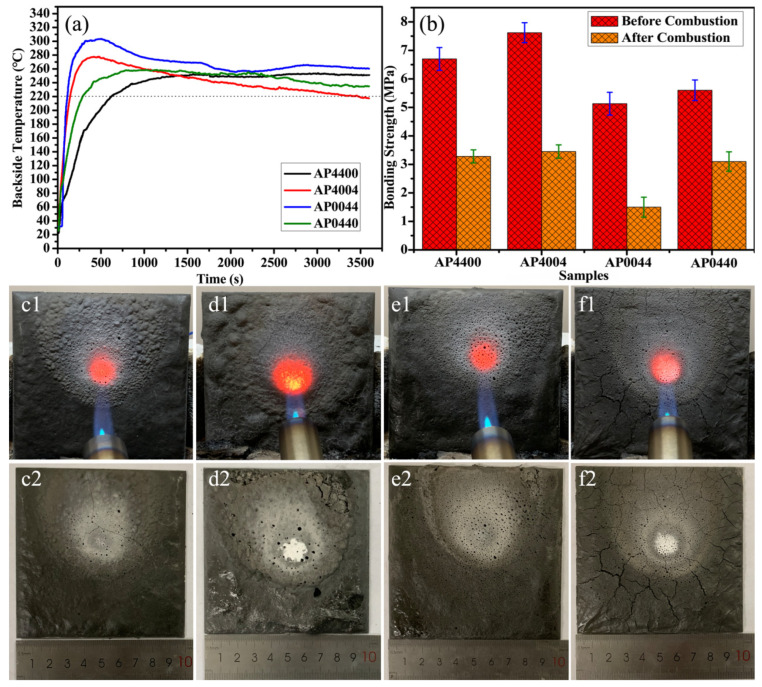
The fire resistance performance of AP4400, AP4004, AP0044, and AP0440: (**a**) the backside temperature under butane flame, (**b**) the comparison of bonding strength before and after combustion, and photos of the coatings (**c1**: AP4400 during combustion, **d1**: AP4004 during combustion, **e1**: AP0044 during combustion, **f1**: AP0440 during combustion, **c2**: AP4400 after combustion, **d2**: AP4004 after combustion, **e2**: AP0044 after combustion, **f2**: AP0440 after combustion).

**Figure 3 materials-16-04498-f003:**
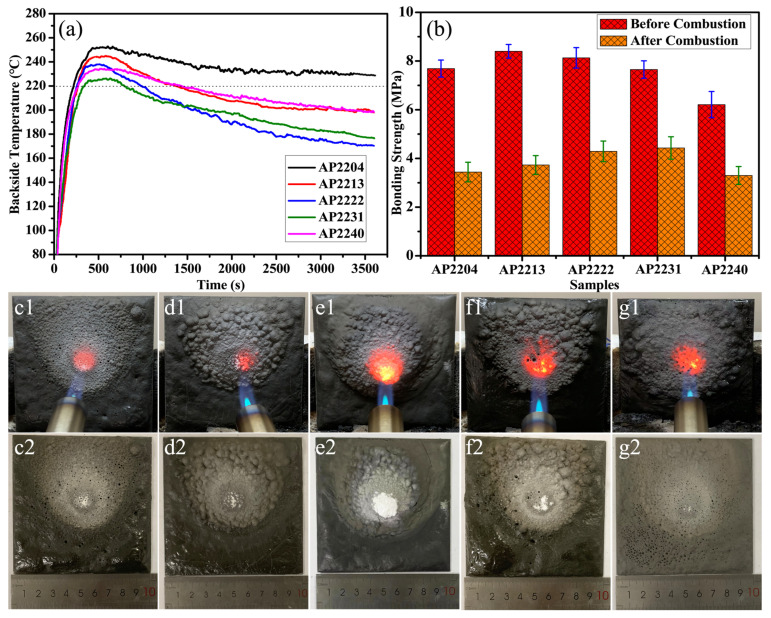
The fire resistance performance of AP2204, AP2213, AP2222, AP2231, and AP2240: (**a**) the backside temperature under butane flame, (**b**) the comparison of bonding strength before and after combustion, and photos of the coatings (**c1**: AP2204 during combustion, **d1**: AP2213 during combustion, **e1**: AP2222 during combustion, **f1**: AP2231 during combustion, **g1**: AP2240 during combustion, **c2**: AP2204 after combustion, **d2**: AP2213 after combustion, **e2**: AP2222 after combustion, **f2**: AP2231 after combustion, **g2**: AP2240 after combustion).

**Figure 4 materials-16-04498-f004:**
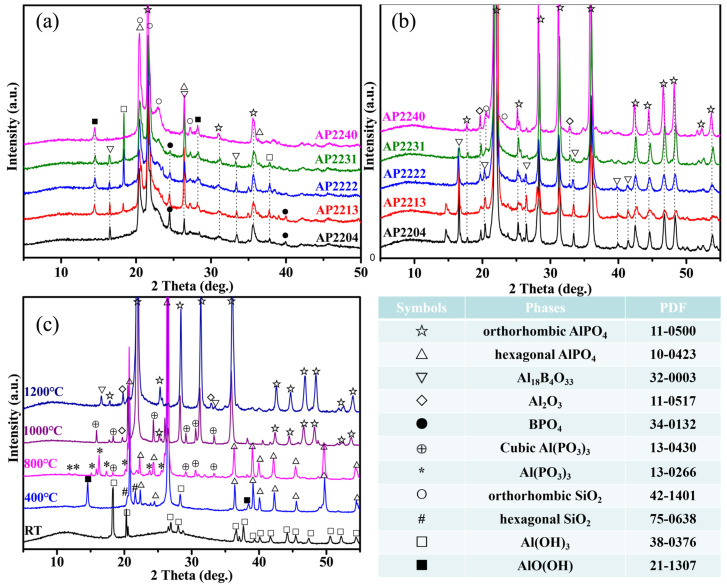
(**a**) XRD spectra of the char of different coatings after 1 h of combustion, (**b**) XRD spectra of various coatings after heat treatment at 1200 °C, (**c**) XRD spectra of AP2231 coating after heat treatment at different temperatures.

**Figure 5 materials-16-04498-f005:**
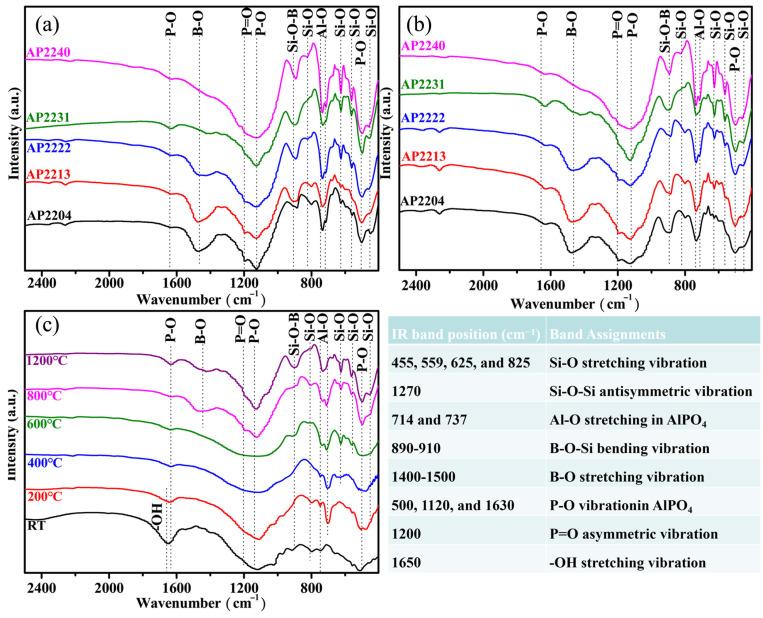
(**a**) IR spectra of the char of different coatings after 1 h of combustion, (**b**) XRD spectra of various coatings after heat treatment at 1200 °C, (**c**) XRD spectra of AP2231 coating after heat treatment at different temperatures.

**Figure 6 materials-16-04498-f006:**
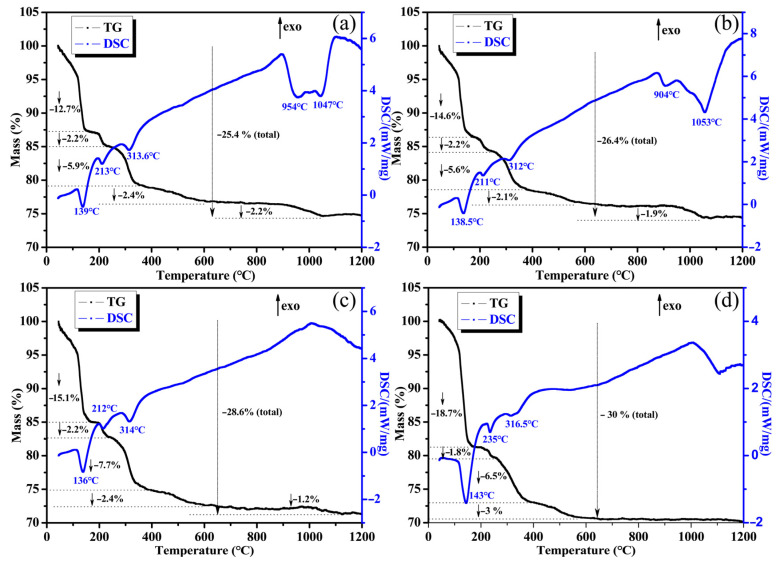
TG-DSC of curves of AP2204 (**a**), AP2222 (**b**), AP2231 (**c**), and AP2240 (**d**) from RT to 1200 °C.

**Figure 7 materials-16-04498-f007:**
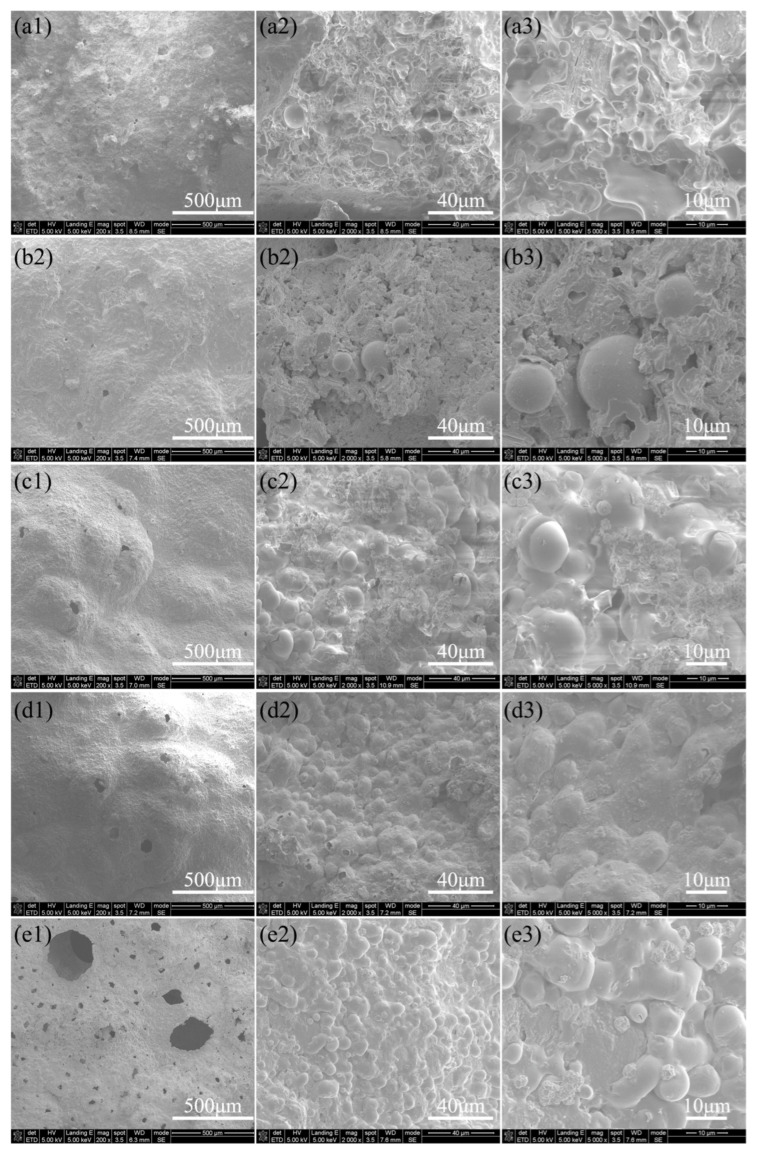
SEM images on the surface of various coatings after combustion (**a1**–**a3**: AP2204, **b1**–**b3**: AP2213, **c1**–**c3**: AP2222, **d1**–**d3**: AP2231, **e1**–**e3**: AP2240; low to high magnification from left to right).

**Figure 8 materials-16-04498-f008:**
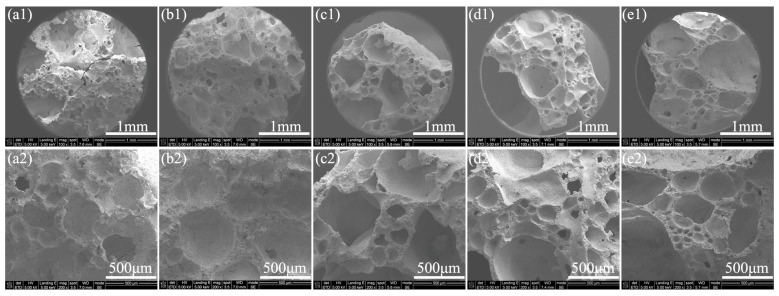
SEM images on the surface of various coatings after combustion (**a1**,**a2**: AP2204, **b1**,**b2**: AP2213, **c1**,**c2**: AP2222, **d1**,**d2**: AP2231, **e1**,**e2**: AP2240; low to high magnification from top to bottom).

**Table 1 materials-16-04498-t001:** Formulas for different coatings.

	Matrix (g)	Nano-Silica Powder (g)	Hollow Silica Beads (g)	Hollow Glass Microspheres (g)	Boron Carbide (g)
AP8000	240	8	0	0	0
AP0800	240	0	8	0	0
AP0080	240	0	0	8	0
AP0008	240	0	0	0	8
AP4400	240	4	4	0	0
AP4004	240	4	0	0	4
AP0044	240	0	0	4	4
AP0440	240	0	4	4	0
AP2204	240	2	2	0	4
AP2213	240	2	2	1	3
AP2222	240	2	2	2	2
AP2231	240	2	2	3	1
AP2240	240	2	2	4	0

## Data Availability

Not applicable.

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
