# Peer review of "A Novel Inorganic Aluminum Phosphate-Based Flame Retardant and Thermal Insulation Coating and Performance Analysis"

_materials, 2023, doi:10.3390/ma16134498_

Round 1
Reviewer 1 Report
We would be interested in reviewing the revised version of the manuscript, once these elements will have been considered.
The title can be also modified and precise but not mandatory to change it.
Reference Number [6] is not added in the text anywhere.
Please associate any errors with the reported values. Meaning of precision and significant figures of values is related to associated errors.
Page 10, lines 31-43 seem a bit misleading after discussion of Fig. Anticipate this paragraph before presenting Figure
English to be revised, especially for vocabulary (e.g. functionality vs functionalization, characters vs characteristics, big and large) and tenses of the verbs.
We would be interested in reviewing the revised version of the manuscript, once these elements will have been considered.
The title can be also modified and precise but not mandatory to change it.
Reference Number [6] is not added in the text anywhere.
Please associate any errors with the reported values. Meaning of precision and significant figures of values is related to associated errors.
Page 10, lines 31-43 seem a bit misleading after discussion of Fig. Anticipate this paragraph before presenting Figure
English to be revised, especially for vocabulary (e.g. functionality vs functionalization, characters vs characteristics, big and large) and tenses of the verbs.
Reviewer 2 Report
Dear Editor,
Thank you for inviting me to review the manuscript entitled "A novel amorphous aluminum phosphate-based flame retardant coating prepared by modifying it with nano silica, hollow silica beads, hollow glass microspheres, and boron carbide" by Cai et al.
From a grammatical and language perspective, the article is of good quality. I suggest that the manuscript be accepted for publication with some revisions, which I have listed below:
In my opinion, the material under study should be referred to as in the first sentence of the abstract instead of as in the title. Additionally, the importance given to the “ratio of glass microspheres to boron carbide” was not made clear, given that other fillers were also added.
Some results were explained in the abstract, such as “AP2231 showing the best fire resistance performance”. However, a brief explanation for other results, such as “AP2231 showing the best bonding performance”, was lacking.
In the Introduction, It would be important to explain the effects of the fillers separately and how they act synergistically. What possible mechanisms led to certain filler proportions achieving better or worse results? It seems that if this blend of three fillers was studied, each of these fillers must have been studied separately. This knowledge should be brought to the introduction.
The introduction fails to fit the proposed work into the current literature. It is unclear whether there are existing studies on inorganic expandable flame retardant coatings incorporated with flame-retardant fillers. A comprehensive literature review should be conducted on this topic, and a summary of that should be included in the introduction to understand if there is any scientific novelty in this article and how the literature can provide insight into the expected results.
Regarding the methodology, the horizontal distance between the sample and the butane flame burner in the flame tests should be reported. I assume it was kept constant throughout the test, but this should be mentioned as well.
As for the curves shown in Figures 1 and 2, is there any dispersion of those points? How many samples were analyzed? The shapes of these curves (Figs. 1 and 2) seem to indicate a growth up to a maximum value and then a decrease, followed by stabilization. This should be better explained in the text.
The conclusion seems appropriate, except for the fact that the same text should not be repeated in the abstract. There appears to be laziness on the part of the authors in dedicating time to each section of the work.
All figures and tables are necessary for the purposes of the manuscript. The figures are of good graphical quality and were well presented. The references used are appropriate and well formatted.
Sincerely,
Reviewer 3 Report
The article is very interesting. Both the research methodology and the results are presented reliably. The analysis of the obtained results requires supplementation and reference to literature data. (The suggestion are presented below). In the summary, the application possibilities of the developed coatings or directions of further research should be indicated. The conclusions are correct.
SUGGESTIONS:
1. Abstract should be corrected. Now, the abstract contain only the key results and main conclusions. The abstract should contain the most important information regarding the context and background of the research, the aim of the article, and investigation methods. The practical application should be summarized. Please add the following information:
· A research gap
· The paper aims
· The test methods
· A practical implication.
2. In the introduction, a very large part of the information concerns fire safety, but too little of the introduction is devoted to the originality of manuscript. The properties of aluminum phosphate are known and presented in the literature already in the years 2000-2005, e.g. :
· M. Vipola, S. Ahmaniemi, et all in Mater. Sci. Eng., A323 (2002), pp. 1-8 presented the aluminum phosphate seal
· B. Formanek, K. SzymaÅ„ski, et all, in Journal of Materials Processing Technology, Volumes 164–165, 2005, Pages 850-855, presented phase composition of a modified aluminium phosphate binder in the temperature range 293–1073 K and properties of sealed coatings
·
|
Temperature (K) |
Phase composition |
|
353 |
Amorphous structure AlH3(PO4)2·3H2O |
|
393 |
Amorphous structure AlH3(PO4)2·3H2O Al(H2PO4)3 hexagonal |
|
473 |
Al(H2PO4)3 hexagonal |
|
523 |
AlPO4 tetragonal, AlH2P3O10·2H2O |
|
683 |
Al(PO3)3 hexagonal, AlH2P3O10 |
|
773 |
Al(PO3)3, Al2P6O18 |
|
873 |
Al(PO3)3, Al2P6O18, AlPO4 tetragonal |
|
1023 |
Rhombic AlPO4, Al(PO3)3 |
Please add two sentences summarizing the literature review, indicating the originality and the novelty of the manuscript.
3. Please kindly specify the information provided in section 3.5.,
"Secondly, the coating has multiple endothermic reactions. With the increase in temperature, multiple endothermic reactions occur in the coating, including the decomposition of phosphate, the decomposition of aluminum hydroxide, the generation of glass and the volatilization of partial boron oxide.
a) What value does the temperature increase in the coating?
4. The analysis of research results requires reference to literature data presented by other authors. Did you obtain better or comparable results? Did the similar mechanisms, e.g. that the liquid B2O3 can repair cracks, or improve the closed pore structure are presented in the literature? What was different?
5. The phosphate coatings presented in the literature did not meet the expectations when applied under the conditions of the synergic effect of high temperature and abrasive wear. Therefore, in the conclusion, application possibilities of the developed coatings or directions of further research should be indicated.
I hope that the above suggestions will allow the reader to better understand the content of the article. The presented study is an important solution for the industry.
